# Pediatric Angioedema without Wheals: How to Guide the Diagnosis

**DOI:** 10.3390/life13041021

**Published:** 2023-04-15

**Authors:** Lucia Liotti, Luca Pecoraro, Carla Mastrorilli, Riccardo Castagnoli, Francesca Saretta, Francesca Mori, Stefania Arasi, Simona Barni, Mattia Giovannini, Lucia Caminiti, Michele Miraglia Del Giudice, Elio Novembre

**Affiliations:** 1Pediatric Unit, Department of Mother and Child Health, Salesi Children’s Hospital, 60123 Ancona, Italy; 2Pediatric Unit, Department of Surgical Sciences, Dentistry, Gynecology and Pediatrics, University of Verona, 37126 Verona, Italy; 3Pediatric Hospital Giovanni XXIII, Pediatric and Emergency Department, AOU Policlinic of Bari, 70126 Bari, Italy; 4Department of Clinical, Surgical, Diagnostic and Pediatric Sciences, University of Pavia, 27100 Pavia, Italy; 5Pediatric Clinic, Fondazione IRCCS Policlinico San Matteo, 27100 Pavia, Italy; 6Pediatric Department, Latisana-Palmanova Hospital, Azienda Sanitaria Universitaria Friuli Centrale, 33100 Udine, Italy; 7Allergy Unit, Meyer Children’s Hospital IRCCS, 50139 Florence, Italy; 8Translational Research in Pediatric Specialties Area, Division of Allergy, Bambino Gesù Children’s Hospital [IRCCS], 00165 Rome, Italy; 9Department of Health Sciences, University of Florence, 50139 Florence, Italy; 10Department of Human Pathology in Adult and Development Age “Gaetano Barresi”, Allergy Unit, Department of Pediatrics, AOU Policlinico Gaetano Martino, 98124 Messina, Italy; 11Department of Woman, Child and General and Specialized Surgery, University of Campania “Luigi Vanvitelli”, 80138 Naples, Italy

**Keywords:** angioedema, idiopathic angioedema, hereditary angioedema, histamine, bradykinin

## Abstract

Angioedema (AE) is a vascular reaction of subcutaneous and submucosal tissues that identifies various clinical pictures and often is associated with wheals. AE without wheals (AEwW) is infrequent. The ability to distinguish between AEwW mediated by mast cells and bradykinin-mediated or leukotriene-mediated pathways is often crucial for a correct diagnostic–therapeutic and follow-up approach. AEwW can be hereditary or acquired. Factors typically correlated with hereditary angioedema (HAE) are a recurrence of episodes, familiarity, association with abdominal pain, onset after trauma or invasive procedures, refractoriness to antiallergic therapy, and lack of pruritus. The acquired forms of AE can present a definite cause based on the anamnesis and diagnostic tests. Still, they can also have an undetermined cause (idiopathic AE), distinguished according to the response to antihistamine in histamine-mediated and non-histamine-mediated forms. Usually, in childhood, AE responds to antihistamines. If AEwW is not responsive to commonly used treatments, it is necessary to consider alternative diagnoses, even for pediatric patients. In general, a correct diagnostic classification allows, in most cases, optimal management of the patient with the prescription of appropriate therapy and the planning of an adequate follow-up.

## 1. Introduction

Angioedema (AE) is a transient vascular reaction of subcutaneous and submucosal tissue that identifies clinical pictures of variable etiology, characterized by an increase in endothelial permeability resulting in tissue swelling in a particular body district [1]. AE is a frequent reason for admission to the emergency department; about 100,000 visits are registered annually in the United States of America [2]. The variability of AEs clinical picture often makes a correct diagnosis and an adequate therapeutic approach difficult. A primary distinction between the forms “with or without wheals” is fundamental in the classification of AE. AE with wheals can be acute or chronic, inducible, or spontaneous [3,4]. The forms of AE without wheals (AEwW) can be linked to various clinical entities. In particular, the hereditary forms are distinguished from the acquired ones [5,6]. Hereditary forms represent an essential chapter that has been well-studied in recent years and has a well-defined diagnosis [7] (Figure 1).

In most cases of acquired AEwW in childhood—the topic of this article—it is possible to identify the triggering cause. When this does not happen, an “idiopathic” AEwW is defined, and these pictures may or may not present a response to antihistamine therapy.

## 2. Epidemiology and Pathogenesis

The exact incidence of pediatric AE is not well known, as it is highly variable according to the association or not of urticaria and according to the specific cause that determined it. In a series of 5918 patients referred to a pediatric allergy center, only 95 cases (1.6%) presented AEwW [9]. Similar percentages were highlighted in a study of 17,823 adults referred to an Argentine allergy center: only 303 patients (1.7%) had a diagnosis of AEwW [10]. AE is characterized by a localized, reversible nonpitting edema involving subcutaneous, submucosal, or deep dermal tissues. It is caused by vasodilation and increased vascular permeability, resulting in extravasation of intravascular fluids [11]. The pathogenic mechanism responsible for AE can be divided into three pathways: mediated by mast cells, mediated by leukotrienes, or mediated by bradykinin (Table 1). According to the response to antihistamines, idiopathic AE can be distinguished into mediated by histamine and not mediated by histamine forms.

Each pathogenic pathway has a unique etiology, immunopathological patterns, and response to treatment. Mast-cell-mediated EA is triggered by the degranulation of mast-cells and basophils, resulting in the release of vasoactive substances (e.g., histamine, prostaglandin D2, leukotriene C4, D4, platelet-activating factor) and the activation of the kinin system. This can occur through a type I hypersensitivity reaction (IgE mediated) following exposure to an allergen or triggered by a direct mast-cell degranulation (pharmacodynamic action caused by, e.g., contrast media, opioid drugs). The most frequently involved allergens are foods, drugs, and insect bites; in this case, the AE may be a spectrum of the allergic reaction. A second mechanism concerns the activation of cysteinyl leukotrienes, which can be induced by non-steroidal anti-inflammatory drugs (NSAIDs). These can cause AE by inhibiting COX-1 [12,14], which determines a shunt of arachidonic acid metabolism and an increase in the production of proinflammatory cysteinyl leukotrienes. In these cases, AE has been associated with the intake of various NSAIDs. In a large retrospective observational study of 1007 atopic children, 4.1% of patients reported facial AE secondary to NSAIDs, with an increasing incidence with age and peaking at 21% in the age group ranging from 16 to 21 years (compared to 2% in patients aged <5 years) [15]. Another pediatric study showed a prevalence of NSAID-induced AEwW in 6% of patients, and NSAID-induced cross-reactivity was reported in all cases [9]. Finally, AE represents the most frequent clinical presentation in children with hypersensitivity to NSAIDs [16]. Therefore, it is essential to investigate the patient's NSAID use during the anamnestic collection. The third pathogenetic mechanism involves an unregulated activation of the kinin system resulting in bradykinin excess. Bradykinin is a vasoactive nonapeptide of the contact system that binds to the transmembrane-expressed kinin B2 receptor on vascular endothelial cells and causes phosphorylation with the destruction of vascular endothelial cadherin (VE-cadherin). VE-cadherin is the key protein involved in the formation of endothelial tight junctions. Its loss results in fluid transfer from the vascular space to the extracellular space, causing an increase in vascular permeability clinically associated with AE [17]. The primary regulator of the contact system is the C1 inhibitor, a serine protease capable of inhibiting a series of enzymes, such as coagulation factors XI and XII, plasma kallikrein, and plasminogen involved in the metabolism of bradykinin. The latter pathway includes hereditary angioedema (HAE), acquired AE caused by C1 esterase inhibitor (C1-INH) deficiency, and AE induced by angiotensin-converting enzyme (ACE) inhibition [18,19]. Interestingly, new genetic mutations causing HAE with normal C1-INH have been identified in recent years, shedding new light on the pathogenesis of these rare forms of HAE [20]. Hormonal factors can also play an important pathogenetic role in AE (e.g., use of contraceptives, mainly estrogens; menstruation; pregnancy; and ovulation), as well as common triggers, such as trauma or stress [21]. In the pediatric age, a triggering factor for AE is represented by infections, particularly viral ones [5]. For example, Herpes simplex, Coxsackie A and B, hepatitis B, Epstein–Barr virus, and other viral diseases, such as upper respiratory tract infections, can produce circulating immune complexes resulting in anaphylatoxins. Immune complexes can activate the complement to release anaphylatoxins C3a, C4a, and C5a; these interact with receptors on mast-cells and basophils and result in mast-cell degranulation with the release of histamine and other vasoactive factors. Bacterial infections are also associated with AE in children, such as otitis media, sinusitis, tonsillitis, upper respiratory tract infections, and urinary tract infections.

## 3. Differential Diagnosis

It is possible to differentiate mast-cell-mediated from bradykinin-mediated and leukotriene-mediated AEwW with a thorough analysis of the clinical and anamnestic characteristics (Table 1). Mast-cell-mediated AE is more likely allergic when associated with hives or itching; however, it can also present without these characteristics when caused by an allergenic trigger. In reports examining patients presenting AEwW, the allergen was identified only in 10–15% of cases among adult and pediatric populations [9,20]. Since the pathophysiology in these cases is IgE mediated, typical triggers are food, drugs, or insect bites, with food as the leading cause in childhood. Aeroallergens have also been reported as a trigger for isolated AE in pediatric patients [9]. Finally, it should be noted that allergy to galactose-α-1,3 galactose (α-gal), which causes delayed signs and symptoms after ingestion of mammalian meat, has been described in a large pediatric cohort, and approximately 1/3 of patients had AE [22]. However, it was not clarified in the cited study whether AE was always associated with urticaria or isolated. AE attacks typically involve multiple regions, including extremities, abdomen, face, oropharynx, or larynx in the bradykinin-mediated form. Attacks in the extremities and the abdomen account for nearly 50% of all attacks in HAE, and at least 50% of patients suffer an upper airway attack with a lifetime risk of asphyxia [18]. Depending on the pathogenesis, episodes can last hours in the mast-cell-mediated form, a variable time in the leukotriene form, and days in the bradykinin-mediated form (Table 1). In the latter, attacks are often severe and disabling and can be associated with significant morbidity and risk of mortality [23]. Other conditions should be mentioned in the differential diagnosis of AEwW. Angioedema with eosinophilia is a rare idiopathic AE that can be classified as episodic or non-episodic. Gleich described the first phenotype in 1984 in patients with recurrent AE, urticaria, fever, weight gain, eosinophilia, and elevated IgM levels [24]. Milder and self-limiting sporadic pictures of peripheral edema, eosinophilia, and arthralgias characterize the second phenotype of AE with eosinophilia. It has been predominantly observed in young Japanese women [25]. Swelling of the lips may be mimicked by Melkersson–Rosenthal syndrome (MRS), a rare neuro-mucocutaneous condition of unknown origin, clinically characterized by a triad of synchronous or metachronous clinical manifestations: recurrent peripheral facial paralysis, recurrent orofacial edema, and fissured tongue [26]. In particular, the edema of the lips is not painful and represents the most frequent sign in the initial stage of this syndrome, also called Meischer MRS or Meischer’s granulomatous cheilitis. The identification of a noncaseating granuloma on a mucocutaneous biopsy suggests the diagnosis. The “idiopathic systemic capillary-leak syndrome” (ISCLS) or Clarkson syndrome should be mentioned in the differential diagnosis [27]. It is a rare, potentially lethal acute condition characterized by recurrent attacks of capillary hyperpermeability and vascular collapse, accompanied by hypoalbuminemia, hemoconcentration, and subcutaneous edema of the face, hands, and hips [20]. Although the pathogenetic mechanism of this syndrome is unknown, a clinical picture dominated by fluid extravasation can complicate pathologies such as septic shock or burns or therapeutic interventions such as treatment with recombinant interleukin-2, bone marrow transplantation, or cardiopulmonary bypass. Other conditions accompanied by AE include hypereosinophilic syndrome, vasculitic urticaria, autoimmune diseases, serum sickness-like reactions (SSLR), and drug rash with eosinophilia and systemic symptoms (DRESS); however, these disorders do not commonly occur with isolated AE and are rare in children [9,19,28]. An “angioedema mimic”, defined as a swelling lasting a few days or weeks, sometimes associated with overlying erythema, has also been described. Defined causes of mimic angioedema include Cushing's syndrome, contact dermatitis, reaction to facial fillers, blepharochalasia, conjunctivitis, Graves' disease, granulomatous cheilitis, superior vena cava syndrome, acne, complex migraine, neurosis, pseudotumor of the orbit [10]. We must also remember that not all swellings are AE, so an accurate and detailed anamnesis, a careful, objective examination, and targeted diagnostic tests are essential for a correct classification, allowing us to exclude pseudo-angioedema forms, which can also occur in children. For example, a pseudo-angioedema may be periorbital cellulitis, which presents as a painful unilateral swelling with overlying skin erythema associated with headache and fever [13]. Another situation that can be defined as pseudo-angioedema is represented by testicular torsion, characterized by an acute painful swelling of the testicle, which can be edematous, that should be determined by ultrasound diagnostics [13]. Finally, it must be remembered that factitious angioedema (a manifestation of a somatoform disorder) can occur in the pediatric population. However, in the work of Feldman and colleagues [29], only four cases were described in adults identifiable with Munchausen syndrome, a rare condition in the child, never described in the form by proxy. This condition should be considered in evaluating the patient with recurrent episodes of AE, particularly in those cases with normal laboratory tests [29].

## 4. Diagnosis

The clinical history and the physical examination are essential for assessing the etiology of AE (Figure 2) [5]. 

The first step is understanding whether AE is associated with urticaria and itching [9]. The following steps are associated with the symptoms’ duration, the swelling’s location, and the presence of AE in the family history. Moreover, a specific investigation of the possible trigger factors, such as food, drugs—in particular NSAIDs—insect bites, exposure to inhalant allergens or physical agents such as pressure or heat, physical exercise, trauma, emotional stress, and alcohol ingestion must be carried out. Specifically, alcohol ingestion can represent a possible cofactor facilitating the development of AE. Patients should be asked about fever and recent viral or bacterial infections. Infections may trigger AE development. In addition, it is essential to deepen the response to drugs, especially antihistamines, and steroids. This aspect is essential in differentiating mast-cell-mediated and bradykinin-mediated AE. Specifically, mast-cell-mediated AE usually responds to antihistamines and steroids, unlike bradykinin-mediated AE. Diagnostic tests must be guided by history and clinical features. If the clinical history suggests it, an allergological evaluation with a skin prick test and dosage of specific IgE for food, inhalants, or Hymenoptera must be performed to identify the etiological agent. When a recurrent AEwW is suspected, these tests are less valuable. In this case, it is important to suspect the presence of hereditary AE. It can be suspected by the presence of many clinical features: isolated recurrent AE, a family history of AE, and abdominal pain associated with trauma as a trigger for AE. When hereditary AE is suspected, testing the C4 dosage is the first step and can usually be performed even in an emergency department setting. Most patients affected by hereditary AE have low C4 levels at baseline, and all patients will have low C4 levels during an AE attack [30]. The next step is to deepen C1 inhibitor (C1INH) activity. Its activity must be less than 50–60% in patients affected by HAE [31]. On the other hand, complement and C1INH levels can be physiologically low in neonates. For this reason, it is usually recommended to postpone the measurement of complement and C1INH levels until one year of age, with the possibility of carrying out genetic tests in case of solid clinical suspicion before one year of life. However, while C4 levels may be primarily influenced by age, the quantitative dosing and functional testing for C1INH may be accurate even in infants [30]. Overall, it may be reasonable to perform these tests even below one year of life. Moreover, genetic mutations causing HAE with normal C1-INH have been reported (Figure 1). When all diagnostic tests are negative, this type of AE is called “idiopathic AEwW”.

## 5. Idiopathic AEwW

Idiopathic AEwW represents a diagnosis of exclusion and a poorly understood entity [8]. Adult studies report idiopathic AE frequency between 33.5% and 46%. [20,32]. In a pediatric study, no etiological factor was found in about 50% of subjects despite a proper diagnostic investigation [9]. Based on the response to pharmacological treatment, there are two types of idiopathic AEwW (Figure 2). The first is responsive to antihistamines (idiopathic mediated by histamine AE). The second is unresponsive to antihistamines (idiopathic non-mediated by histamine AE). This specific type of AE can be differentiated according to the response to higher doses of H1-antihistamines (up to 4-fold standard dose), omalizumab, and tranexamic acid [8,33]. In a systematic review, several treatment options were distinguished for patients with AE refractory to conservative treatments [34]. In particular, idiopathic non-histamine-mediated AE showed efficacy of omalizumab in 63% of cases with no further attacks after starting treatment. In addition, in idiopathic AE, tranexamic acid improved symptoms in 73% of cases and resulted in a complete absence of symptoms in 16% of cases.

### 5.1. Idiopathic Histamine-Mediated AEwW

Histaminergic AE is generally the most frequent form of AE without wheals. It is divided into acute and chronic depending on the duration of the signs and symptoms (acute < 6 weeks, chronic > 6 weeks) [35]. This term describes patients with recurrent episodes of AE without urticaria, whose symptoms are controlled by H1-antihistamine treatment. A specific cause cannot be identified in these patients [36]. Given the recent identification, its prevalence is not well known. 87% of patients responded to antihistamine therapy in a study of 294 patients with idiopathic AEwW followed over ten years. [20] This group of patients typically is characterized by an episode of AE that developed rapidly, reached a maximum expression within 6 h, and resolved within 24 h. The most common localizations are represented by the face, extremities, and gastrointestinal (GI) tract in 30% of cases. Although present in patients of all ages, it occurs preferentially between 36 and 42 years of age. Omalizumab has been used successfully in some subjects who have failed to respond to maximal antihistamine therapy [8].

### 5.2. Idiopathic Non-Histamine-Mediated AEwW

This type of AE identifies an unfamiliar, non-hereditary form of AE characterized by the absence of specific causes and the absence of a response to antihistamine therapy. It usually occurs between the ages of 20 and 40 and affects males in particular. It is mainly characterized by facial edema, present in almost all patients, and tends to resolve in less than 48 h. In addition, it can frequently recur [6]. In some cases, it responds to tranexamic acid therapy. Long-term prophylaxis is suggested in subjects with >2 severe attacks per month [20,33,37].

## 6. Pediatric Features of Acquired or Idiopathic Histamine-Mediated AEwW

Little is known about the prevalence, clinical presentation, etiology, management, and follow-up of pediatric-acquired or idiopathic isolated histamine-mediated AE. To the best of our knowledge, only the prospective study (95 patients) by Ertoy Karagol et al. and the retrospective study (42 patients) by Ocak et al. deepen this theme in the pediatric age [9,38]. Specifically, these studies included subjects affected by histamine-mediated AEwW, including in their cohorts patients affected by AE with poor response to antihistamines, the simultaneous presence of urticaria, or positive family history of AE [9,38]. Cross-sectional studies on pediatric populations identified a prevalence of 1.6% of isolated AE [9]. A majority was seen in males (42.9–71.6%), and the average onset of the first episode of pediatric-acquired isolated histamine-mediated AE was 7–7.8 years. [9,38] The first visit to a pediatric allergy unit was made at 8–9 years old. The mean duration of the AE exacerbation episode was 24 h; 2–5 episodes of isolated AE exacerbations were reported in the previous 12 months for each patient. Only 23.2% of patients experienced a single exacerbation episode. The most frequent localizations were represented in eyelids and lips, followed by extremities, genitals, ears, and tongue. On the other hand, the legs and abdomen, representing common body areas of HAE exacerbations, were not involved; 62.2% of patients had multiple exacerbation areas, and 11–45.3% had a history of atopic disease. The most common comorbidity was allergic rhinitis. Only 11.9% of patients reported the presence of at least one family member with episodes of AE during their lives. This prevalence data equals 91.2% in pediatric HAE [9,38]. Regarding the etiology, 51% were affected by episodes of AE of unknown cause (pediatric–idiopathic isolated histamine-mediated AE) [9]. When identified (pediatric-acquired isolated histamine-mediated AE), the most frequent etiology was represented by infectious factors, followed by allergic diseases, hypersensitivity to NSAIDs, and thyroid diseases. The infectious etiology is prevalent among the known causes of isolated AE, sharing this feature with acute urticaria in children [39]. Allergic conjunctivitis and insect bites are the most prevalent causes of allergic disease [9]. Compared to HAE, the low prevalence of a traumatic event as a trigger factor for AE exacerbation is notable [38]. Regarding the prophylaxis of episodes of idiopathic histamine-mediated AEwW, a three-month trial with the antihistamine desloratadine may represent an excellent therapeutic option. Thyroid disease may represent a risk factor linked to the prolongation of prophylaxis with antihistamine therapy [9]. The prognosis seems good in the short-term follow-up. The exacerbation of isolated AE after cessation of antihistamine prophylaxis appears infrequent. In the study by Okak et al., 81% of patients accessed the Emergency Department on at least one exacerbation episode; 9.4% of patients were hospitalized due to an AE exacerbation episode in at least one event [38]. The most significant features for the characterization of the pediatric patient affected by pediatric-acquired or idiopathic isolated histamine-mediated AE seems to be related to gender, age of onset, age at diagnosis, the average duration of the exacerbation episode, sites involved, trigger factors for exacerbation, personal and family history, and response to antihistamine therapy. These features are summarized in Table 2.

## 7. Conclusions

The term “AE” includes clinical manifestations shared by different illnesses, often associated with urticaria. When AE is not associated with urticaria, it can be related to other manifestations. The ability to distinguish among AE mediated by mast cells or bradykinin or leukotriene pathways is often crucial for a correct approach and follow-up. Specifically, AEwW can be hereditary or acquired. The presence of recurrent episodes and family history, the association with abdominal pain, the onset after trauma or invasive procedures, the absence of response to antihistamine therapy, and the lack of pruritus are all factors typically correlated with forms of HAE [38]. On the other hand, the acquired forms of AE can have a definite cause based on the anamnesis and diagnostic tests or an undetermined cause (idiopathic forms). When the reason is unknown, it is essential to distinguish the forms of AE mediated or not mediated by histamine according to the response to antihistamines. The most common form of AE in children responds to antihistamine treatment. If the AE does not respond to the commonly used treatment, it is necessary to consider alternative diagnoses and rare illnesses that can present with AE, even in pediatric patients. A correct diagnostic workup generally allows optimal patient management by prescribing appropriate therapy and planning an adequate follow-up.

## Figures and Tables

**Figure 1 life-13-01021-f001:**
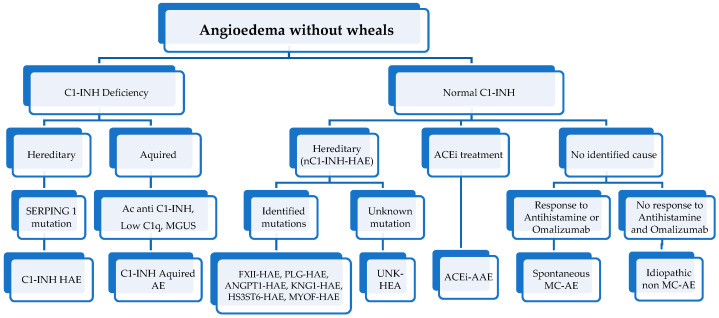
Classification of angioedema without wheals (modified from [1,8]) (HAE, hereditary angioedema; AAE, acquired angioedema; MGUS, monoclonal gammopathy of unknown signification; ACEi, angiotensin converting enzyme inhibitors; FXII, Hageman factor gene; PLG, plasminogen gene; ANGPT1, angiopoietin 1 gene; KNG1, Kininogen 1 gene; MYOF, myoferlin; HS3ST6, heparan sulfate-glucosamine 3-O-sulfotransferase 6; UNK, unknown mutations; MC, mast-cell).

**Figure 2 life-13-01021-f002:**
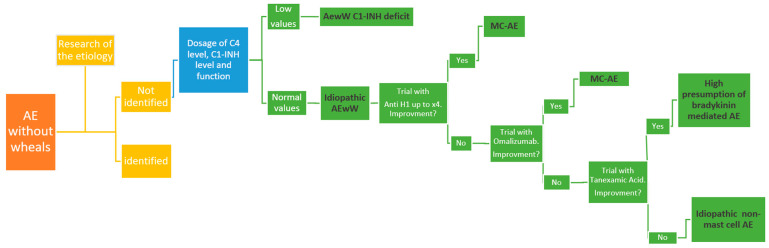
Diagnostic and therapeutic approaches to AEwW (modified from [8]).

**Table 1 life-13-01021-t001:** Characteristics of different types of AEwW (modified from [12,13]).

	Types of Angioedema		
	Mediated by Mast Cells	Mediated by Bradykinin	Mediated by Leukotrienes
**Pathogenetic Mechanism**	• IgE-mediated response to antigen exposure with the release of vasoactive mediators • Non-IgE-mediated (may be associated with chronic spontaneous urticaria) • Idiopathic (unidentifiable cause)	Complex interaction of complement, coagulation, and contact system • Hereditary angioedema with C1-INH deficiency or defect • Hereditary angioedema with normal C1-INH • Acquired angioedema with C1-INH deficiency • ACE inhibitor-induced angioedema	Inhibition of cyclooxygenase-1 determines a shunt of the arachidonic acid metabolism and causes an increase in 5-lipoxygenase activity • Induced by NSAIDs, aspirin
**Response to antihistamine in 12 h**	Yes	No	
**Urticaria**	Frequent	Absent	
**Age of onset**	Anyway	Often in the 1^st^ or 2^nd^ decade (40% within 5 years)	Anyway
**Pruritus**	Present	Slight	
**Duration of edema**	Usually <48 h	Often >72 h	Variable
**Preferred localizations**	Face (eyelids, lips), neck	Face, abdomen, extremities	Periorbital, airways
**Prodromal symptoms**	No	Often	
**Trauma as a trigger**	No	Yes	
**Development of signs and symptoms**	Fast	Slower	Hours
**Familiar history**	Never	Often	Not known
**Further investigations**	Levels of serum tryptase (useful in the context of anaphylaxis/allergenic trigger)	Serum C4 Quantitative and qualitative levels of C1-INH (all normal in ACE inhibitor-induced angioedema)	

**Table 2 life-13-01021-t002:** Characteristics of the patients affected by acquired or idiopathic histamine-mediated AEwW in pediatric age [9,38].

Demographic, Personal History and Clinical Data	
Gender	Male
Average age at first-episode onset	7–7.8 yrs
Average age at diagnosis	8–9 yrs
Average length of relapse episode	24 h
Number of exacerbation episodes in the last year	2–5
Most involved body areas	Eyelids, lips
Number of body areas involved during exacerbation episodes	Several
Personal history	Atopy
Familial history	Absence of family history related to recurrent AE episodes
Trigger factors	Infections, respiratory or food allergens, thyroid diseases, drugs (medical history of trauma is not frequent)
Response to antihistamine treatment	Symptoms improvement
Need for adrenaline or intubation	No
Prognosis	Good

## Data Availability

Not applicable.

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
