# Peer review of "Pediatric Angioedema without Wheals: How to Guide the Diagnosis"

_life, 2023, doi:10.3390/life13041021_

Round 1

Reviewer 1 Report

The authors aimed to provide a guide for diagnosis and characterization of angioedema in pediatric patients. In my opinion, this “Perspective Article” does not add anything new to the existing knowledge about angioedema syndrome and quite the opposite is the case: The article uses unusual terminology and very old references and it even confuses readers by making false statements. Therefore, I do not recommend publication of the article in this form. I can support my decision with at least 12 arguments:

1.       The first sentence of the article gives a definition about angioedema, but there is no reference added to this sentence. To me, it does not make sense to explain angioedema by using edema. Better to use “swelling”. Furthermore, I miss that angioedema is paroxysmal and localized event.

2.       Why do the authors use the term angioedema without urticaria. Of course, I know what the authors want to say it is important and very relevant to know, that the word urticaria is not a symptom but a disease/condition. Urticaria is defined as the occurrence of wheals and/or angioedema The authors cite this international guideline in Ref. 2). In other words, exclusively recurrent angioedema can also be urticaria by definition. Most urticaria patients have both, wheals and angioedema, others have only wheals and only a very few have recurrent angioedema only. Therefore, using the terminology angioedema without urticaria is very confusing to readers and not established. I guess, the authors mean Angioedema without wheals. If so, than they should use this term.

3.       Figure 1 shows an adapted Figure from Cicardi et al 2014. This is a very old reference. Obviously, the authors are not aware of that many things changed the world of recurrent angioedema in the past 8-9 years. For example, far more mutations in the HAE with normal C1-inhibitor have been identified, e.g. KNG1, Myoferlin, Angiopoetin etc….The authors only show HAE-FXII. This is insufficient and disinformes the reader. I recommend the authors read the international guideline for Management of HAE: Maurer M, Magerl M, Ansotegui I, Aygören-Pürsün E, Betschel S, Bork K, Bowen T, Balle Boysen H, Farkas H, Grumach AS, Hide M, Katelaris C, Lockey R, Longhurst H, Lumry WR, Martinez-Saguer I, Moldovan D, Nast A, Pawankar R, Potter P, Riedl M, Ritchie B, Rosenwasser L, Sánchez-Borges M, Zhi Y, Zuraw B, Craig T. The international WAO/EAACI guideline for the management of hereditary angioedema-The 2017 revision and update. Allergy. 2018 Aug;73(8):1575-1596. doi: 10.1111/all.13384. Epub 2018 Mar 12. PMID: 29318628.

4.       Line 54-57: This is misinformation! I think the authors do not really know what acquired C1INH deficiency (AAE-C1INH) really means. Again, I strongly recommend the authors to read the guideline stated above. Moreover the authors state that patients with C1INH deficiency (acquired) may respond to antihistamines. WHAT? This is not true.  

5.       Line 64: Why do the authors repeat a definition of angioedema as they already did in the first sentence? However, this definition sounds better and the authors also present a reference.

6.       Lines 66-69. It is correct, that the most frequently found types of angioedema are the mast-cell mediated and the bradykinin mediated angioedema. I recommend to avoid the names histaminergic and bradykininergic as those are not well-established. Moreover, the authors should consider that there are many other causes of angioedema, i.e. drug induced due to ACE inhibitors (most likely also BK mediated) or due to endothelial dysfunction)

7.       This sentence is very confusing: Histaminergic angioedema without urticaria (AEwU) can also occur in  …….simultaneous manifestation of chronic spontaneous urticaria.

8.       The statement about Melkersson Rosenthal syndrome in line 147 is not correct. I understand that the authors want to discuss differential diagnosis but why do they state that these patients have angioedema? It is a Cheilitis granulomatosa those patients have not angioedema.

9.       Figure 2. Why do the authors only recommend to test C4 and C1INH function, but not C1INH levels? This is not correct. All three should be checked according to the international guideline for HAE

10.   I can not understand why the authors recommend tranexamic acid when patients do not respond to antihistamines (standard dose? 4-fold dose?). This is not correct. The next step would be introducing omalizumab, which is approved for urticaria at year 12 since 2014. Now I see, that the authors use a reference from 2006.

11.   Line 195: The authors actually want to focus on pediatric patients and here the write about alcohol. I really miss the focus on pediatric patients in that article.

12.   Idiopathic histaminergic AEwU is a not-established terminology…

Author Response

Dear reviewer,

We thank you for the opportunity offered to revise our article.

We have considered your suggestions, and we have made all the appropriate and suggested changes.

We have substantially revised the entire manuscript, according to your requests. 

Reviewer 2 Report

The authors review the very complicated topic of the angioedema without urticaria in pediatric population. It seems that they have exhausted the current literature and present a useful paper to the practicing clinicians.

The contribution of genotyping in the diagnosis must be presented.

The language of the paper must be polished

Author Response

Dear reviewer,

We thank you for the opportunity offered to revise our article.

We have considered your suggestions, and we have made all the appropriate and suggested changes.

Reviewer 3 Report

The article focuses on pediatric forms of angioedema without urticaria. The article faces important aspects of this often underestimated condition, such as differential diagnosis and pediatric features of bradykinin-mediated angioedema. I think the paper is interesting and well-written. I have only a few suggestions that can further increase the quality of the manuscript.

Introduction:

Lines 51-52 and Figure 1: Recently, the awareness of the pathophysiology of HAE has increased. In particular, other genetic mutations in patients with HAE with normal C1-INH have been identified, produced by mutations in other genes besides C1-INH and FXII (e.g., plasminogen (PLG-HAE), angiopoietin-1 (ANGPT1-HAE), kininogen-1 (KNG1-HAE), myoferlin (HAE-Myoferlin), and HS3ST6 gene). Please mention them in the introduction and consider adding the following reference Clin Rev Allergy Immunol. 2021 Jun;60(3):305-315. doi: 10.1007/s12016-021-08835-8. 

Table 1, third column: 

- line 1: please replace the word "contact" with "contact system".

- lines 3-6: I think lines 4,5 and 6 are a repetition since FXII, Ang1, and PLG cause hereditary angioedema with normal C1-INH, which has already been reported in line 3. In addition, other mutations are involved in these HAE forms, as reported in my previous comment. I would suggest removing the specific forms and only keeping the line with "hereditary angioedema with normal C1-INH" or including all mutations known up to date by adding kininogen-1, myoferlin, and HS3ST6.

Please check the abbreviations throughout the manuscript.

Author Response

(The authors gave the same response as above.)
